# Caregivers’ Perceived Emotional and Feeding Responsiveness toward Preschool Children: Associations and Paths of Influence

**DOI:** 10.3390/nu13041334

**Published:** 2021-04-17

**Authors:** Carla Fernandes, Ana F. Santos, Marilia Fernandes, António J. Santos, Kelly Bost, Manuela Verissimo

**Affiliations:** 1William James Center for Research, ISPA-Instituto Universitário, 1149-041 Lisboa, Portugal; csfernandes@ispa.pt (C.F.); ana.f.santos2@hotmail.com (A.F.S.); mfernandes@ispa.pt (M.F.); asantos@ispa.pt (A.J.S.); 2Human Development and Family Studies, University of Illinois, Champaign, IL 61801, USA; kbost@illinois.edu

**Keywords:** parental responsiveness 1, emotion regulation 2, feeding practices 3

## Abstract

Although there is a large body of research connecting emotion to eating behaviors, little is known about the role of caregivers’ responses to children’s emotions in the context of child feeding. The purpose of this study was to analyze the relation between caregivers’ emotional responsiveness and feeding responsiveness. The mothers of 137 children between 2 and 6 years of age reported on their responses to children’s negative emotions using the Coping with Children’s Negative Emotions Scale and on their feeding practices using the Comprehensive Feeding Practices Questionnaire. The results showed that mothers’ supportive emotion responses (e.g., problem-focused, emotion-focused, and expressive encouragement reactions) tend to be positively associated with responsive feeding practices (e.g., encouraging, modelling, and teaching healthy food-related behaviors). Instead, mothers’ unsupportive responses (e.g., distress, punitive and minimization reactions) tend to be positively associated with nonresponsive feeding practices (e.g., food as reward or to regulate emotions, and pressure to eat) and negatively associated with responsive feeding practices. Our results suggest that emotional and feeding responsiveness may be intertwined and that differences in parent’s emotional responsiveness may translate into differences in their feeding styles, setting the stage for parents’ use of positive vs. negative feeding practices.

## 1. Introduction

Parents are powerful socialization agents of children’s emotions and eating behavior. They influence children’s self-regulation of emotions through their emotion regulation strategies [1] and their self-regulation of energy intake through their feeding practices [2]. Looking at how parents support children’s emotion regulation could provide insights about the developmental processes by which caregivers shape children’s self-regulation of intake [3]. Difficulties at this level could translate into children’s less effective emotion regulation (i.e., emotion dysregulation) and self-regulation of energy intake, which represent risk factors for childhood obesity [4,5,6,7,8,9,10]. Thus, gathering evidence about how emotion responsiveness and feeding responsiveness are linked has also critical implications for pediatric obesity.

Pediatric obesity is a major health and social challenge, and developmental models highlight specific processes (e.g., emotion regulation) that might be related to the emergence of healthful and unhealthful eating patterns. In Europe, the prevalence of childhood obesity has reached alarming proportions [11,12,13], and despite the recent decreasing tendency, Portugal remains one of the European countries with the highest prevalence of children with overweight (29.6%) and obesity (12.0%) [14,15]. It is important to look at this information, considering the complex multifactorial status of childhood obesity and viewing it as the result of an interaction between multiple risk factors [16,17]. Beyond biological influences, factors within the family’s obesogenic environments need to be taken into account, including socialization influences on children’s emotion regulation and self-regulation of energy intake. Despite empirical and practical relevance, only a few empirical studies addressed the relation between responses to children’s emotions and feeding in the parenting context.

The ability to successfully regulate energy intake is related to how responsive parents’ feeding practices are to children’s cues of hunger and satiety [18,19,20,21,22]. Responsive feeding is characterized by the caregiver’s ability to recognize and respond promptly and appropriately to the child’s cues of hunger and satiety, as well as setting up predictable routines (e.g., offer food in a predictable schedule so the child is likely to be hungry) and provide a variety of foods in order for the child to get the nutrients needed to grow healthy [20,21,23]. On the other hand, non-responsive feeding includes parental coercive and excessive controlling feeding practices (e.g., pressuring, rewarding, monitoring or restricting the child’s food intake). These types of parental responses ignore the child’s internal cues of hunger and satiety and instead provide external cues [19,21,24,25]. Research has shown that such feeding practices can place children at risk for excessive weight gain [26,27,28].

This view of responsive feeding derives from an overall framework of developmental concepts like responsive parenting and parental sensitivity [23,29,30]. Responsive parenting has been studied for several years, especially in the context of emotions (i.e., responsiveness to children’s emotions) [31,32,33,34] and has been associated with positive outcomes including emotion regulation, cognitive and language development, secure attachment and self-regulation (e.g., executive function, inhibitory control) [35,36,37,38,39]. However, emotional responsiveness has remained outside the context of child feeding research, which is surprising since there is a substantial amount of empirical evidence connecting emotion to eating behaviors [4,5,6,7].

In early childhood, caregivers play a crucial role in shaping children’s emotion regulation development [40,41,42,43,44]. Particularly, caregivers’ responsiveness to children’s emotions is considered a primary method of emotion socialization [31] with important implications for children’s social-emotional adjustment and development of healthy relationships and behaviors [45,46,47,48,49]. Caregivers’ emotional responsiveness is inextricably linked to the quality of caregiver–child relationships [50,51,52,53]. Especially, the quality of parent–child attachments has important implications for understanding the link between children’s self-regulation of emotions and self-regulation of energy intake. At this level, insecure attachment has been associated with children’s unhealthy food consumption indirectly through unsupportive responses to children’s distress [54]. In fact, a small body of research suggests that emotional responsiveness parallels feeding responsiveness [3,20,25].

According to Frankel and colleagues [3], emotions are eminently present in family meals when it comes to feeding young children. Caregivers present children new flavors, tastes, and textures which can often lead to tensions and battles between the caregiver and his/her child and elicit negative emotions like frustration, anger, and sadness that in turn have an impact on the quality of family meal times. Children tend to be reluctant or refuse new foods, which is believed to be a biological predisposition or an evolutionary-based response that has evolved due to its adaptive value in protecting humans from potentially poisonous foods [55]. That being said, it comes to caregivers to frequently decide how to deal with their children’s reluctance or refusal to eat food [56]. For example, if the caregiver responds with anger to the child’s refusal to eat vegetables, this response may increase the emotionally negative quality of vegetables for the child. Identically, if sweets are associated with happy events such as birthday parties, this may increase the emotionally positive quality of sweets [3]. Additionally, if caregivers respond to the child’s negative emotions around food in a minimizing, distressed, or punitive way, it is possible that this response may have a negative impact in both self-regulation of emotions and energy intake of the child. An example is when the child is crying because he/she is full and the caregiver is pressuring him/her to clean the plate and ignores his/her distress, pressuring the child to eat or even making a threat (e.g., “stop crying and clean your plate or I’ll give you something to cry about”). These kinds of parental responses may reflect a lack of validation and acceptance of the child’s emotions and will not help the child to deal with negative emotions in an adaptive way [1,3,57]. Instead, it can contribute to undermining the child’s ability to respond to hunger and satiety cues [2,26,58,59]. Therefore, parental responsiveness to children’s negative emotions around food may have an impact on children’s energy intake.

Although it is possible that parental influences on children’s self-regulation of emotions and on children’s self-regulation of energy intake may be intertwined, very few studies have examined this link [3]. To our knowledge only three studies present empirical evidence at this level: two of them, focusing on clinical samples, with eating disorders [25,60], and only one on a normative sample, with typical eating behavior [54]. This information appears integrated in mediation models designed to explore influences of other caregivers’ variables (e.g., parents’ binge eating; adult attachment style) on feeding practices and children’s obesity-related outcomes (e.g., child weight, food consumption) through their influence on emotion regulation [25,54,60]. These three studies shared common findings regarding how parents’ negative emotion regulation responses to their children’s negative emotions may serve as risk factors for their use of non-responsive/unhealthful feeding practices.

Thus, the aim of the current study was to analyze how caregivers’ emotional responsiveness and feeding responsiveness are related. At a more specific level, we explored associations between important caregiver’s emotion regulation responses to child’s negative emotions and their food-related practices, reported by mothers. After, we tested the relation of positive and negative emotion regulation strategies to responsive/healthful and non-responsive/unhealthful feeding styles. We expected that the caregiver’s positive emotion regulation responses to child’s distress would be associated with feeding practices that match with a more responsive feeding style, whereas negative emotion regulation responses would be associated with an unhealthful feeding style that reflects the use of non-responsive feeding practices.

## 2. Materials and Methods

### 2.1. Participants

Participants were mothers of 137 children (64 girls and 73 boys) between 2 and 6 years old (*M* = 46.65 months; *SD* = 14.53). Mothers’ ages ranged from 23 to 49 years (*M* = 35.05; *SD* = 5.03) and fathers’ ages from 25 to 59 years (*M* = 38.42; *SD* = 5.89). Mothers’ education level varied between 9 and 21 years (*M* = 16.22; *SD* = 3.04) and fathers’ between 5 and 21 years (*M* = 15.13; *SD* = 3.64). Most parents worked full-time (94.0% mothers; 96.9% fathers), and the large majority were married or cohabiting (89.7%).

Both parents of 2- to 6-year-old children, from 7 schools in a metropolitan area of central-southern Portugal were invited to participate in the study, although only mothers responded. Informed consent was obtained from all the participants. This study was approved by the Ethics Committee (I/038/06/2020).

### 2.2. Measures

#### 2.2.1. Parental Responsiveness to Children’s Negative Emotions

Parents’ responses to children’s negative emotions were assessed using the Coping with Children’s Negative Emotions Scale (CCNES) [61]. CCNES includes 12 hypothetical scenarios in which the child is emotionally upset (e.g., “If my child becomes angry because he/she is sick or hurt and can’t go to his/her friend’s birthday party, I would …”). For each scenario, six possible responses are provided (e.g., “Comfort my child and try to make him/her feel better”; “Tell my child that he/she is overreacting”). Using a 7-point rating scale, parents were asked to rate how likely they were to react in each of six specific ways (1 = very unlikely; 7 = very likely). These six qualitatively different responses to the child’s negative emotional expressions correspond to six subscales: *problem-focused reactions* (α = 0.78) that help the child to solve the problem that caused him/her distress; *emotion-focused reactions* (α = 0.81), which reflect the strategies that help the child feel better; *expressive encouragement* (α = 0.91) that promotes the child’s negative emotional expressions and reflect the parents’ acceptance of those expressions; *minimization reactions* (α = 0.73) that reflect parents’ devaluation of the child’s problem or negative emotional expression; *punitive reactions* (α = 0.67), which involve parents’ use of verbal or physical punishment to control the child’s negative emotional expression; *distress reactions* (α = 0.53), which reflect parents’ discomfort with the child’s negative emotional expression [62]. As proposed by Bost and colleagues [54], two composite scales were also considered. One that reflects total positive emotion regulation strategies (average of *expressive encouragement*, *emotion-focused reactions*, and *problem focused reactions* subscales), and the other reflecting negative emotion regulation strategies (average of *distress*, *punitive*, and *minimization* subscales).

#### 2.2.2. Parental Feeding Practices

Parental feeding practices were assessed using the Comprehensive Feeding Practices Questionnaire (CFPQ [63]). CFPQ consists of 49 items. For each item, using a 5-point rating scale, parents were asked to indicate the degree to which they agreed with a statement (1 = disagree, 2 = slightly disagree, 3 = neutral, 4 = slightly agree, 5 = agree; e.g., “I encourage my child to try new foods”) or how often they use a feeding strategy (1 = never, 2 = rarely, 3 = sometimes, 4 = mostly, 5 = always; e.g., “How much do you keep track of the high-fat foods that your child eats?”). Items comprise 12 subscales that correspond to specific feeding practices: *child control* (α = 0.54) assesses the degree to which parents allow the child to control their own eating behaviors and parents–child feeding interactions; *emotion regulation* (α = 0.88) refers to parents’ use of food to regulate the child’s emotions; *encourage balance and variety* (α = 0.59) refers to parents’ promotion of healthy and varied food consumption; *environment* (α = 0.64) assesses the degree to which parents make healthy foods available in the house; *food as reward* (α = 0.57) involves parents’ use of food as a reward for child’s behavior; *involvement* (α = 0.61) assesses the degree to which parents encourage child’s involvement in meal planning and food preparation; *modelling* (α = 0.77) describes parents being enthusiastic about and actively demonstrating healthy eating for the child; *monitoring* (α = 0.86) refers to parents keeping track of child’s intake of unhealthy foods; *pressure* (α = 0.70) describes parents encouraging the child to eat more food at meals, disregarding child’s satiety/hunger; *restriction for health* (α = 0.63) refers to parental control of child’s intake in order to limit unhealthy foods; *restriction for weight* (α = 0.74) refers to parental control of child’s intake in order to maintain or decrease the child’s weight; *teaching about nutrition* (α = 0.63) describes the degree to which parents use explicit didactic techniques to encourage the child’s intake of healthy foods. As proposed by Bost and colleagues [54], two composite scales were also considered. One reflects emotion-related, pressuring feeding styles (average of the food as a reward, emotion regulation, and pressure), and the other reflects healthy feeding styles (average of the *teaching about nutrition*, *modelling*, *involvement*, *balance and variety*, and *environment*).

### 2.3. Statistical Analysis

Initial analyses explored descriptive statistics as well as the relations between variables (i.e., parental responsiveness to children’s negative emotions and parental feeding practices) and demographics. Child sex and age, as well as parents’ education level, were included as control variables in relevant analyses. A MANOVA was used to predict the effect of parents’ reactions (both positive and negative strategies) on their feeding practices (both healthy and pressuring styles), followed by two ANOVAs (one for healthy and other for pressuring feeding styles).

Because missing data were present for some participants, Little’s MCAR statistic was computed (χ^2^ = 1076.74, *p* = 0.99 for CCNES and χ^2^ = 836.37, *p* = 0.69 for CFPQ), and estimation maximization (EM) algorithm used to imput missing data.

## 3. Results

Descriptive statistics are presented in Table 1. Mothers reported using more positive than negative emotion reaction strategies (*M* = 5.81, *SD* = 0.70 and *M* = 2.55, *SD* = 0.55 respectively; *t* (136) = 38.81, *p* < 0.001), as expected in a non-clinical sample as ours. The same happened regarding healthy and pressuring feeding styles (*M* = 4.04, *SD* = 0.46 and *M* = 2.15, *SD* = 0.52, respectively; *t* (136) = 29.66, *p* < 0.001).

Regarding sex differences, only one CFPQ scale—*teaching about nutrition—*reached significance, with mothers of girls presenting significantly higher scores than mothers of boys (*M* = 3.97, *SD* = 0.75 and *M* = 3.59, *SD* = 0.90, respectively; *F* = 7.17, *p* < 0.01); no other significant differences were found. Child age was not significantly associated with any dimension of CCNES. However, regarding CFPQ, we found that as children were older, mothers reported using more *involvement* and *teaching about nutrition* strategies (*r* = 0.24, *p* < 0.001 and *r* = 0.21, *p* < 0.01, respectively).

When we analyzed parents’ social demographics associations with CCNES and CFPQ, we found that older mothers tended no to use *food as reward* (*r* = −0.22, *p* < 0.001); there was no other association regarding mothers’ or fathers’ age. Regarding parents’ education, higher levels were significantly associated with less CCNES *emotion-focused reactions* for both parents (mother *r* = −0.35, *p* < 0.001 and father *r* = −0.20, *p* < 0.001). Mothers’ education was also related to less *problem-focused reactions* as well as to the *positive reaction composite* (*r* = −0.27, *p* < 0.01 and *r* = −0.24, *p* < 0.001, respectively). When analyzing CFPQ, mothers’ higher education levels were associated with higher *environment* and *monitoring* (*r* = 0.21, *p* < 0.05 and *r* = 0.19, *p* < 0.05, respectively) whereas fathers’ higher education levels were associated with higher *modelling* and *balance and variety* (*r* = 0.19, *p* < 0.05 and *r* = 0.21, *p* < 0.05, respectively).

In the following analyses, we examined associations between the way parents react to children’s negative emotions and their feeding practices, controlling for child sex and age, as well as for both parents’ education level (see Table 2). When analyzing the negative composite of parents reactions, we could see that the more the parents used negative reactions, the less they promoted healthy and varied food consumption (*r* = −0.25, *p* < 0.01) or made it available in the house (*r* = −0.29, *p* < 0.01); the less they were enthusiastic about demonstrating healthy eating (*r* = −0.33, *p* < 0.001), the more they used food to regulate their child’ emotions (*r* = 0.26, *p* < 0.01) or as a reward (*r* = 0.24, *p* < 0.01) and the more they encouraged to eat more at meals, disregarding child’s satiety/hunger (*r* = 0.32, *p* < 0.001). Analyzing the positive composite of parents’ reactions, we found that the more the parents used positive reactions, the more they promoted healthy and varied food consumption (*r* = 0.48, *p* < 0.001) and the more enthusiastic they were about demonstrating healthy eating (*r* = 0.32, *p* < 0.001) or using didactic techniques to encourage the child’s intake of healthy foods (*r* = 0.41, *p* < 0.001). Finally, parents who reported using more negative reactions strategies also reported adopting more pressuring and less healthy feeding styles (*r* = 0.39, *p* < 0.001 and *r* = −0.31, *p* < 0.001, respectively). Parents who reported using more positive reactions strategies reported adopting more healthy feeding styles (*r* = 0.49, *p* < 0.001).

A MANOVA analysis was performed in order to predict the effect of parents’ reactions (both positive and negative strategies) on feeding practices (both healthy and pressuring styles). Child age and sex as well as parents’ education level were added as covariates. Using Pillai’s trace, there was a significant effect of both positive and negative reactions (*V* = 0.20, *F* (2,102) = 12.85, *p* < 0.001 and *V* = 0.18, *F* (2,102) = 10.92, *p* < 0.001). Two ANOVAs revealed significant effects of positive reactions on healthy feeding styles (*F*(1,108) = 18.87, *p* < 0.001) and significant effects of negative reactions on both healthy (*F*(1,108) = 6.71, *p* < 0.01) and pressuring feeding styles and (*F*(1,108) = 16.97, *p* < 0.001). Specifically, negative reactions were found to increase pressuring (*B* = 0.37, *t* = 4.12, *p* < 0.001) and decrease healthy feeding styles (*B* = −0.19, *t* = −2.59, *p* < 0.01), and positive reactions to increase healthy feeding styles (*B* = 0.25, *t* = 4.34, *p* < 0.001).

## 4. Discussion

Our results show that, when mothers reported a higher use of positive responses to their children’s distress (i.e., through helping the child to solve the problem that caused him/her distress, to feel better, and by accepting and promoting his or her negative emotional expressions), they also reported a higher use of responsive feeding practices that reflected the promotion of healthy and varied food consumption, the demonstration of healthy eating, and the encouragement of the child’s intake of healthy foods through the use of explicit didactic techniques. Mothers who reported a higher use of problem-focused reactions (helping the child to solve the problem) also reported encouraging more the child’s involvement in meal planning and food preparation. In turn, mothers’ negative responses to children’s negative emotion—either distress, punitive, or minimization reactions (reflecting discomfort with the child’s negative emotional expression, the use of verbal or physical punishment to control the child’s negative emotional expression, and devaluation of the child’s problem or negative emotional expression, respectively)—appeared associated with higher levels of encouraging the child to eat more food at meals, disregarding child’s satiety/hunger. Additionally, mothers who reported more punitive reactions also reported to use food as a reward for child’s behavior or to regulate the child’s emotions, and mothers who reported more distress reactions also reported lower levels of promotion of healthy and varied food consumption and demonstration of healthy eating for the child. These results are in line with Saltzman and colleagues’ [45] study (the only one where we found evidence regarding the associations between all CCNES and CFPQ dimensions). These findings are consistent with the notion that emotion responsiveness and feeding responsiveness are linked processes, as suggested in previous studies [3,25,54,60].

Mothers’ use of positive and negative emotion regulation strategies (reflected on their likelihood to mobilize supportive and non-supportive emotional responses to deal with children’s distress, respectively) might predict their healthful and unhealthful feeding styles (reflected on their reported use of responsive and non-responsive feeding practices, respectively). As hypothesized, our results indicated that the caregiver’s positive emotion regulation responses to child’s distress predicted the increased use of responsive feeding practices (e.g., promoting balance and variety; modelling and teaching about nutrition) that reflect teaching and modelling feeding styles, thus also predicting healthy feeding styles (opposed results were found when they responded more negatively). In contrast, caregiver’s negative emotion regulation strategies predicted the increased use of non-responsive feeding practices (e.g., food to regulate emotion or as a reward and pressure to eat) that reflect emotion-related, pressuring feeding styles, thus predicting unhealthy or obesogenic feeding styles. These results corroborate previous findings, highlighting the importance of maternal emotion responsiveness for understanding feeding styles that emerge in interpersonal contexts of feeding behaviors [3,25,54,60].

Taken together, the results of this study add to the extant small body of research suggesting that emotional and feeding responsiveness may be intertwined and that differences in parent’s emotional responsiveness may translate into differences in their feeding styles, by setting the stage for parents’ use of positive vs. negative feeding practices [25,54,60]. Thus, these findings are consistent with previous empirical evidence and add to the current state of knowledge by exploring these links among mothers with more typical eating behaviors and by providing evidence regarding supportive emotion responses, beyond unsupportive emotion responses when examining their influence on caregiver’s feeding styles.

We recognize some limitations that constrain the generalization of these results. For instance, we only used maternal self-report measures of both constructs, and our data are correlational and concurrent. This precludes any causal inferences, so caution is needed when interpreting regression results. In addition, like the large majority of studies, this study focused only on parents’ responsiveness to children’s negative emotions, leaving out the equally important responsiveness toward positive emotions [64]. In this sense, it would be important for future studies to also take into account parental responsiveness to children’s positive emotions. Also, we did not control for caregiver’s eating disorders and child’s weight. It would have been important to include these variables, since previous studies have shown their influence in parental feeding practices (e.g., [25,65]).

Future research would benefit from using longitudinal designs to test paths of influence that not only address unidirectional effects but also consider the possibility for bidirectional relationships between children’s emotion regulation and parental feeding practices over time [3]. When possible, future studies should also adopt a multi-informant (e.g., including both mothers and fathers) and multi-method approach. For example, it will be important to include observational data of children’s dietary intake and parent-child interactions surrounding food, namely, parent–child emotion regulation strategies [3,54]. Additionally, given the lack of heterogeneity of the present sample, it would be important to replicate these findings in larger non-convenience and more diverse samples.

Regardless of these limitations, our findings contribute to the current state of knowledge about how responses to children’s negative emotion are related to feeding in the parenting context, translating into the use of feeding practices to regulate children’s energy intake. These findings have research and practical implications, highlighting the need for transdisciplinary work. More research is needed in order to identify mechanisms through which emotion regulation may translate into feeding regulation, possibly taking adult and child attachment into account. For instance, Bost and colleagues [54] found that attachment insecurity was indirectly linked to children’s unhealthful food consumption through unsupportive responses to negative emotion. Gathering this type of data is important to broaden our understanding of the links between feelings and food. We know that caregivers play a crucial role in children’s emotion regulation and eating, two important aspects for healthy development that have implications for children overweight and obesity [25,54]. There is evidence linking emotion dysregulation and eating self-regulation to weight-related outcomes in children [3,17,66,67,68]. For example, parents’ use of negative feeding practices may contribute to the development of unhealthy eating behaviors [27,28,54,69]; also, non-responsive feeding practices and unsupportive responses to negative emotion predict child body mass index [20,25], placing children at risk of weight gain and obesity. The prevalence of childhood obesity and associated comorbidities, along with its major impact on economy and society, attests to the importance of studying family factors embedded in children’s obesogenic environments. Studying the links between caregivers’ emotion regulation strategies (emotional responsiveness) and their feeding styles (feeding responsiveness) could help to unveil potentially modifiable protective and risk factors.

## 5. Conclusions

Our results show that mothers’ positive responses to their children’s distress relate to a higher use of responsive feeding practices (e.g., promoting more healthy and varied food consumption, demonstrating healthy eating, or using didactic techniques to encourage the child’s intake of healthy foods). In turn, when mothers respond more negatively (e.g., through distress, punitive, and minimization reactions) to their children’s distress, they also use more non-responsive or negative feeding practices (e.g., food to regulate emotions or as a reward and pressure to eat more) and less positive feeding practices. In that sense, our findings suggest that emotional and feeding responsiveness may be related, with differences in emotional responsiveness likely reflecting the use of positive vs. negative feeding practices.

## Figures and Tables

**Table 1 nutrients-13-01334-t001:** Descriptive Statistics for the Coping with Children’s Negative Emotions Scale and Comprehensive Feeding Practices Questionnaire.

		Total Sample		Boys		Girls	
		*M (SD)*		*M (SD)*		*M (SD)*	
CCNES	Problem-focused reactions	6.08	(0.70)	6.14	(0.75)	6.01	(0.63)
	Emotion-focused reactions	6.04	(0.73)	6.10	(0.75)	5.96	(0.70)
	Expressive encouragement	5.31	(1.12)	5.33	(1.17)	5.29	(1.07)
	Minimization reactions	2.94	(0.79)	2.92	(0.80)	2.96	(0.79)
	Punitive reactions	2.03	(0.63)	1.99	(0.62)	2.08	(0.64)
	Distress Reactions	2.68	(0.62)	2.64	(0.61)	2.72	(0.64)
	Positive reaction strategies	5.81	(0.70)	5.86	(0.74)	5.75	(0.67)
	Negative reaction strategies	2.55	(0.55)	2.52	(0.54)	2.59	(0.55)
CFPQ	Child control	2.37	(0.58)	2.34	(0.58)	2.41	(0.58)
	Emotion Regulation	1.32	(0.49)	1.27	(0.46)	1.39	(0.51)
	Balance and variety	4.63	(0.42)	4.62	(0.47)	4.64	(0.36)
	Environment	4.29	(0.56)	4.28	(0.58)	4.29	(0.55)
	Food as Reward	2.27	(0.87)	2.16	(0.78)	2.40	(0.96)
	Involvement	3.43	(0.91)	3.39	(0.93)	3.47	(0.88)
	Modelling	4.08	(0.76)	4.03	(0.79)	4.14	(0.73)
	Monitoring	4.45	(0.68)	4.51	(0.72)	4.38	(0.65)
	Pressure	2.86	(0.90)	2.84	(0.91)	2.88	(0.90)
	Restriction for health	2.46	(0.83)	2.36	(0.86)	2.58	(0.78)
	Restriction for weight	2.12	(0.67)	2.09	(0.64)	2.14	(0.69)
	Teaching about nutrition	3.77	(0.85)	3.59	(0.90)	3.97	(0.75)
	Pressuring Feeding Styles	2.15	(0.52)	2.09	(0.47)	2.22	(0.56)
	Healthy Feeding Styles	4.04	(0.46)	3.98	(0.46)	4.10	(0.44)

M = Mean; SD = Standard Deviation; CCNES = Coping with Children’s Negative Emotions Scale; and CPFQ = Comprehensive Feeding Practices Questionnaire.

**Table 2 nutrients-13-01334-t002:** Correlation between *Coping with Children’s Negative Emotions Scale* and *Comprehensive Feeding Practices Questionnaire* dimensions and composites, controlling for child sex and age and parents’ education level.

		**CCNES**							
		**DR**	**PN**	**MI**	**EE**	**EF**	**PF**	**PRS**	**NRS**
**CFPQ**	CC	0.10	0.04	−0.02	0.04	0.24 *	0.13	0.14	0.04
	ER	0.18	0.31 ***	0.15	−0.11	−0.07	−0.10	−0.12	0.26 **
	BV	−0.26 **	−0.36 ***	−0.10	0.43 ***	0.34 ***	0.42 ***	0.48 ***	−0.29 **
	E	−0.13	−0.26 **	−0.21 *	0.11	−0.07	0.07	0.06	−0.25 **
	FR	0.17	0.32 ***	0.10	−0.07	0.02	−0.07	−0.06	0.24 **
	I	−0.15	−0.07	0.04	0.13	0.16	0.21 *	0.19 *	−0.06
	MD	−0.32 ***	−0.30 ***	−0.19 *	0.27 **	0.25 **	0.30 **	0.32 ***	−0.33 ***
	MN	−0.07	−0.05	−0.08	0.17	0.12	−0.02	0.13	−0.08
	PR	0.25 **	0.23 *	0.29 **	−0.19 *	0.08	−0.11	−0.11	0.32 ***
	RW	0.03	0.15	0.09	−0.09	−0.05	−0.06	−0.09	0.11
	RH	0.14	0.18	0.06	0.03	0.02	−0.12	−0.02	0.15
	TN	−0.20 *	−0.15	−0.10	0.39 ***	0.21 *	0.39 ***	0.41 ***	−0.18
	PFS	0.29 **	0.40 ***	0.26 **	−0.18	0.03	−0.13	−0.13	0.39 ***
	HFS	−0.31 ***	−0.30 ***	−0.15	0.39 ***	0.26 **	0.41 ***	0.43 ***	−0.31 ***

* *p* < 0.05; ** *p* < 0.01; *** *p* < 0.001; CPFQ: CC = Child Control, ER = Emotion Regulation, BV = Balance and Variety, E = Environment, FR = Food as Reward, I = Involvement, MD = Modelling, MN = Monitoring, PR = Pressure, RH = Restriction for Health, RW = Restriction for Weight, TN = Teaching about Nutrition, PFS = Pressuring Feeding Styles Composite, and HFS = Healthy Feeding Styles Composite; CCNES: DR = Distress Reactions, PN = Punitive Reactions, MI = Minimization Reactions, EE = Expressive Encouragement, EF = Emotion-Focused Reactions, PF = Problem-Focused Reactions, PRS = Positive Reactions Composite, and NRS = Negative Reaction Composite.

## Data Availability

The raw data supporting the conclusions of this article will be made available by the authors, without undue reservation.

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
