# Peer review of "Caregivers’ Perceived Emotional and Feeding Responsiveness toward Preschool Children: Associations and Paths of Influence"

_nutrients, 2021, doi:10.3390/nu13041334_

Round 1
Reviewer 1 Report
I would like to thank the authors for preparing an interesting work about Caregivers ’perceived emotional and feeding responsiveness 2 toward preschool children. The topic is very valuable. There are some issues to discuss. Introduction The authors should specify the percentage of obese children in Europe and Portugal (please give the specific indicators) with the current source. The authors mention about the aims of the study several times in the text, for example they re-refer once again to the goal of the work in the sub-chapter "Feeding responsiveness and emotional responsiveness". The aims of the study should be specified only once in the text, preferably at the end of the section introduction. The authors added two subchapters in the introduction section (1.1 and 1.2), I would prefer to move these subchapters to the discussion section), you can keep the structure of subchapters but add the other subchapters with the text that is currently the content of the discussion. Subchapters 1.1. and 1.2 are very valuable but I advice to shorten their length so that they are not the main part of the discussion. Material and methods Section "Participants" needs to be described in more detail. The authors write that the participants of the study were parents (and mother and father) - As a reader I don't know do you take into analysis the child when f.ex. only the mother agreed for a study, Did both parents fill in the questionnaire? Or was it enough that only one parent answered the questions? Please specify precisely the inclusion criteria. What were the inclusion criteria for children? Only age? Or maybe other factors, e.g. lack of chronic diseases? Was the child's weight and BMI taken into account? It would be worthwhile to analyze in this context as well. Results Table 1. Please add a legend explaining the abbreviations M and SD Table 2- is not readable, I would suggest not to use the abbreviations DR PN ... but their full names Please add a separate chapter "Conclusions", in which the authors should briefly summarize the obtained results and they should relate to the aims of the study.Author Response
Please see the attachment.

Reviewer 2 Report
see the attached file

Round 2
Reviewer 1 Report
I would like to thank the Authors for improved version of the manuscript, I accept it in the current version.